# Eosin Y-Functionalized Upconverting Nanoparticles: Nanophotosensitizers and Deep Tissue Bioimaging Agents for Simultaneous Therapeutic and Diagnostic Applications

**DOI:** 10.3390/cancers15010102

**Published:** 2022-12-23

**Authors:** Gabriel López-Peña, Silvia Simón-Fuente, Dirk H. Ortgies, María Ángeles Moliné, Emma Martín Rodríguez, Francisco Sanz-Rodríguez, María Ribagorda

**Affiliations:** 1Departamento de Física Aplicada, Universidad Autónoma de Madrid, C/Francisco Tomás y Valiente 7, 28049 Madrid, Spain; 2Departamento de Química Orgánica, Universidad Autónoma de Madrid, C/Francisco Tomás y Valiente 7, 28049 Madrid, Spain; 3Departamento de Física de Materiales, Universidad Autónoma de Madrid, C/Francisco Tomás y Valiente 7, 28049 Madrid, Spain; 4Nanomaterials for Bioimaging Group, Instituto Ramón y Cajal de Investigación Sanitaria, 28034 Madrid, Spain; 5Institute for Advanced Research in Chemical Sciences (IAdChem), Universidad Autónoma de Madrid, 28049 Madrid, Spain; 6Departamento de Biología, Universidad Autónoma de Madrid, C/Darwin 2, 28049 Madrid, Spain

**Keywords:** photodynamic therapy, upconversion nanoparticles, infrared imaging, eosin Y, ROS

## Abstract

**Simple Summary:**

This work describes a nanoplatform for deep-tissue photodynamic therapy (PDT) and imaging using upconverting nanoparticles functionalized with eosin Y (EY), a photosensitizer (PS) that have been demonstrated to be effective even in hypoxic environments. These structures take advantage of the capability of the nanoparticles to be excited by 800 nm near infrared light, where penetration is higher in comparison with visible light commonly used in PDT, and the thermal load is minimum. Additionally, the combination with UCNPs enables the transport of EY into the cell, which is not possible for EY alone. The generation of reactive oxygen species (ROS) under 800 nm light inside the cell becomes therefore possible based on upconversion and energy transfer processes. These UCNPs also present long lasting infrared fluorescence under 808 nm excitation, thus enabling their use as deep tissue bioimaging agent during PDT.

**Abstract:**

Functionalized upconverting nanoparticles (UCNPs) are promising theragnostic nanomaterials for simultaneous therapeutic and diagnostic purposes. We present two types of non-toxic eosin Y (EY) nanoconjugates derived from UCNPs as novel nanophotosensitizers (nano-PS) and deep-tissue bioimaging agents employing light at 800 nm. This excitation wavelength ensures minimum cell damage, since the absorption of water is negligible, and increases tissue penetration, enhancing the specificity of the photodynamic treatment (PDT). These UCNPs are uniquely qualified to fulfil three important roles: as nanocarriers, as energy-transfer materials, and as contrast agents. First, the UCNPs enable the transport of EY across the cell membrane of living HeLa cells that would not be possible otherwise. This cellular internalization facilitates the use of such EY-functionalized UCNPs as nano-PS and allows the generation of reactive oxygen species (ROS) under 800 nm light inside the cell. This becomes possible due to the upconversion and energy transfer processes within the UCNPs, circumventing the excitation of EY by green light, which is incompatible with deep tissue applications. Moreover, the functionalized UCNPs present deep tissue NIR-II fluorescence under 808 nm excitation, thus demonstrating their potential as bioimaging agents in the NIR-II biological window.

## 1. Introduction

Rare-earth-doped nanoparticles (RENPs) possess fascinating optical properties, which have triggered their use in a wide range of fields, from pH or temperature sensors to imaging and sensing [1,2]. The interest in these nanostructures originates from their chemical stability, their low toxicity, and their ability to produce narrow, differentiable peaks with a strong emission in several spectral regions, especially in the near-infrared (NIR) range of the electromagnetic spectrum with its transparency windows [3]. These windows (also referred to as biological windows) range from 680–950 nm (NIR-I) and 1000–1350 nm (NIR-IIa) and allow researchers to achieve unique penetration of light into tissues [4,5,6]. Taking advantage of these characteristics helped to increase the biological applications of RENPs for non-invasive real-time imaging of cells and tissues [7]. Moreover, one of the most outstanding properties of RENPs is their ability to convert infrared light into visible light by a process known as photon upconversion (UC), which has opened-up a new range of in vivo applications, especially in deep-tissues. One of the most striking possibilities of upconversion nanoparticles (UCNPs) is their combination with photosensitizers to enable deep-tissue NIR excited photodynamic therapy (PDT) [8,9]. PDT consists in the generation of cytotoxic reactive oxygen species (ROS) by means of a photosensitizer (PS) and light [10]. ROS generated close to or within tumor cells causes irreversible damage that ultimately leads to cell death. PDT is a widely applied technique to treat cancer because it does not require complex instrumentation; it is selective and has fewer side effects for patients than other techniques, like minimal non-tumoral tissue toxicity or invasiveness [11]. However, due to the attenuation effect of light through the tissues (skin, blood, etc.), PDT is only used to treat small tumors on or just under the skin, or on the lining of internal organs or cavities. Moreover, many PS suffer from several drawbacks, such as a tendency to aggregate in biological media, which leads to poor cellular uptake, and their low-to-moderate ROS generation due to the poor light penetration [3]. Indirect ROS production, stemming from the energy transfer between photosensitizers and NPs, is a ground-breaking method for PDT as it solves the issue of poor light penetration and allows PDT in deep tissues (even at more than 1 cm depth). Therefore, several examples based on the combination of UCNPs and PS have appeared in recent years, most of them based on the use of 980 nm-excited UCNPs [12]. Although studies have shown that photoinduced damage to cells is minimal at this wavelength, the use of 980 nm light interferes with the absorption band of water, causing a decrease in light penetration and resulting in undesired heating, and therefore, thermal damage to surrounding healthy tissue. The alternative is the use of UCNPs excitable at 800 nm, where minimum cell damage is present and the absorption of water is negligible, which increases tissue penetration and improves the specificity of treatment [13].

There are few studies that employ PDT mediated by UCNPs under 800 nm excitation to obtain intense visible emissions [14,15,16,17]. However, their excellent deep tissue imaging capabilities and capacity to act as nanocarriers, allowing internalization of ROS-generating molecules otherwise unable to cross the cell membrane, remain to be explored in PDT studies.

An example of molecules falling into the latter class is eosin Y (2-(2,4,5,7-tetrabromo-6-oxido-3-oxo-3H-xanthen-9-yl)benzoate) (hereafter EY). EY is an inexpensive commercially available pH-sensitive dye that is traditionally used as a biological stain in histology and as a food colorant. EY is also an interesting photoactive compound used as an organic photoredox catalyst. Furthermore, EY has been reported to show higher ROS production efficiency under oxygen-poor conditions (hypoxia) when irradiated with green light [18]. As most tumors are essentially hypoxic, EY is an excellent candidate for efficient PDT in vivo. However, a major limitation is the inability of EY to cross living cell membranes, since it can only pass through partially broken cell membranes (fixed cells), and therefore, its biological applications as a photosensitizer are drastically limited [19,20]. To solve this problem, we envisioned that the attachment of EY to RENPs could circumvent the problem of cellular uptake due to the good biocompatibility of NPs and reported endocytosis [21]. Furthermore, this would allow the exploitation of EY as PS using 808 nm excitation due to the extraordinary photophysical properties of UCNPs (Figure 1A and Appendix A).

Here, we report the study of rare-earth doped NPs decorated with EY as a NIR-excitable photosensitizer for PDT and bioimaging. These decorated UCNPs were easily prepared by a simple surface functionalization of UCNPs with EY. We performed biological evaluations, including cellular uptake and toxicity assays, and evaluated the ability of eosin-UCNPs as novel photosensitizer for PDT and its capability to produce ROS under NIR-light irradiation. Finally, the potential of UCNPs to simultaneously act as contrast agent for intensity-based and fluorescence lifetime-based deep-tissue imaging was also evaluated.

## 2. Materials and Methods

### 2.1. Synthesis of Rare-Earth-Doped Nanoparticles

Sodium gadolinium fluoride nanoparticles, triply doped with neodymium, ytterbium, and thulium (NaGdF_4_: Nd^3+^, Yb^3+^, Tm^3+^ NPs) were synthesized using a procedure previously reported [22]. Briefly, 1 mmol of the corresponding RECl_3_ were added to a 100 mL flask together with 10 mL of oleic acid and 10 mL of 1-octadecene and heated to 120 °C under vacuum for 1 h. After letting the solution cool down to room temperature, 2.5 mmol of sodium trifluoroacetate were added and the temperature of the solution was maintained at 300 °C under N_2_ for 2 h. The solution was taken up in 250 mL centrifuge tubes and the UCNPs were precipitated using absolute ethanol, isolated via centrifugation (6000 rpm, 20 min), and washed twice with a 1:5 hexane/ethanol mixture. Subsequently, the oleate was removed following a procedure from the literature [23]. Briefly, the UCNPs were dispersed in water (10 mg/mL) and the pH was reduced to 3–4 by adding 1 M HCl dropwise. The mix was shaken overnight and then transferred to a separatory funnel. The aqueous phase was washed three times with 5 mL of dichloromethane (DCM). The combined organic phases were extracted once with 5 mL of water and the combined aqueous phase was washed once more with DCM. After separation, the aqueous phase was centrifuged (6000 rpm, 20 min) and ligand-free UCNPs were obtained.

### 2.2. Synthesis of the NaGdF_4_: Nd^3+^,Tm^3+^,Yb^3+^@eosin Y NPs (EY-UCNPs)

Ligand-free UCNPs (21 mg) and an excess of eosin Y (100 mg) were dispersed by sonication in 4 mL H_2_O. The solution was brought to a pH of 8–9 by adding 1 M NaOH dropwise and stirred overnight. The EY-UCNPs were recovered by centrifugation (6000 rpm, 20 min) and washed five times with 2 mL of H_2_O.

### 2.3. Synthesis of NaGdF_4_: Nd^3+^,Tm^3+^,Yb^3+^@PEG-eosin Y NPs (EY-PEG-UCNPs)

Ligand-free UCNPs (21 mg, 0.16 mmol) and hetero-bifunctional polyethylenglycol (PEG) amino-PEG12-carboxylic acid (CA-PEG, 10 mg, 0.016 mmol) were stirred in 4 mL of PBS overnight. This was followed by centrifugation, two washes with PBS, and redispersion in 4 mL PBS. In a second flask, 10.4 mg of eosin Y (0.016 mmol) was dissolved in 1 mL of 50 mM MES buffer and 2.8 mg NHS (0.02 mmol) followed by 3.7 mg of EDC (0.02 mmol) was added and the mixture, which was stirred for 2 h before being added to the suspension of the PEG-UCNPs. Subsequently, the stirring was continued overnight and EY-PEG-UCNPs were obtained by centrifugation, followed by washing four times with water to remove unreacted eosin. A reaction scheme towards the final products can be found in Appendix A.

### 2.4. Transmission Electron Microscopy

Transmission electron microscopy (TEM) images were obtained in a JEOL JEM1010 microscope (JEOL USA, Inc., Peabody, MA, USA) operating at 100 KV. The sample was prepared by dropping sample dispersions (1 mg/mL in ethanol) onto a 300-mesh carbon coated copper grid (3 mm in diameter) followed by evaporation of the solvent.

### 2.5. Dynamic Light Scattering

Dynamic Light Scattering (DLS) measurements were performed with a Zetasizer Nano ZS instrument (Malvern Panalytical, Malvern, UK) using a 0.1 mg/mL dispersion of NPs in dimethyl sulfoxide (DMSO) contained in a standard 1 cm quartz cuvette on. The energy source was a red laser (630 nm wavelength), and the angle between the sample and the detector was 173 °C.

### 2.6. Fourier Transform Infrared (FTIR) Spectroscopy

Spectra of the as synthesized (oleate-capped), ligand-free, and eosin-capped NPs were measured with an ATR-FTIR module of the Cary 630 FTIR (Agilent, Santa Clara, CA, USA) by depositing a droplet of the dispersion on the diamond cell and letting the solvent evaporate.

### 2.7. Emission and Lifetime Spectroscopy

Fluorescence spectra of the NPs in the visible light were measured using a 790 nm fibre coupled laser diode and a short-pass filter blocking all wavelengths above 750 nm. The collected emission was spectrally analyzed using a monochromator (iHR320, HORIBA, Kyoto, Japan) and recorded using a cooled CCD array detector (Synapse, HORIBA).

Fluorescence characterization of the NPs in the NIR was performed using a 790 nm fiber-coupled laser diode and a long-pass filter blocking all wavelengths below 850 nm. The fluorescence was then spectrally analyzed by a Shamrock 193i compact imaging spectrograph (0.21 nm of spectral resolution) and a 1.7 µm InGaAs IDus CCD detector (Model Number DU490A, both from Andor Technology, Belfast, UK).

For fluorescence lifetime measurements the sample was excited with an optical parametric oscillator (OPO, Lotis TII LT-2214, Minsk, Belarus) pumped by the third harmonic of a 10 ns Nd:YAG Q-Switched laser. The OPO provides 10 ns pulses, with 4 mJ of pulse energy and a 10 Hz repetition rate. The emission wavelength was selected by a Shamrock 193i compact spectrograph, and then detected using an IR photomultiplier (Hamamatsu Photonics, Shizuoka, Japan). The decay curves of the emitted signals were measured with a 500 MHz digital oscilloscope (Lecroy Waverunner LT372, Chestnut Ridge, NY, USA).

### 2.8. Detection of ROS Generation

ROS generation was evaluated by irradiating a quartz cuvette containing a water/DMSO 1:1.5 dispersion of the NPs with a 790 nm laser diode by monitoring the quenching of the chemical trap 1,3-diphenylisobenzofuran (DPBF). The total volume of solution used was 2.5 mL with a concentration of 0.1 mg/mL of DPBF, and a concentration of NPs of 0.4 mg/mL. The whole setup was kept in darkness and at constant temperature by using a CUV-QPOD temperature-controlled sample compartment system from Ocean Optics to avoid degradation of the DPBF.

### 2.9. Cell Viability Tests

Cell viability of HeLa cells exposed to the EY-UCNPs and/or to NIR light was analyzed by the MTT (3-[4,5-dimethylthiazol-2-yl]-2,5-diphenyltetrazoliumbromide) colorimetric assay [24]. This method is based on the capacity of living cells which possess dehydrogenases to reduce the tetrazolium salt MTT to a colored and insoluble compound, formazan. In the first experiment, HeLa cells were incubated for 2 h in the dark with the corresponding NPs. After 24 h, cells were incubated with MTT (0.1 mg/mL in DMEM with 10% SFB and and 50 units/mL penicillin and 50 µg/mL streptomycin) for 2 h at 37 °C. Then the MTT-containing medium was removed and crystallized formazan was suspended in DMSO. Afterwards, we proceeded to measure the absorbance at 540 nm using a plate reader (Synergy HTX Multi- Mode Microplate Reader (BioTek^®^, Santa Clara, CA, USA)). Cell viability was estimated as a percentage relative (100% viability) to the mean of the absorption obtained from the control cells (not incubated with NPs). To observe the effect of infrared light irradiation on culture viability, after 2 h of incubation with the different nanoconjugates, cells were exposed to NIR irradiation at different times (using a Hydrosun^®^750 lamp as source, Müllheim, Germany). Twenty-four hours later, cell viability was analyzed using the MTT method. A colored filter was used to block the red emissions from the lamp (see spectra in Appendix A).

### 2.10. Fluorescence Microscopy Imaging

Microscopic observations and photographs were performed in an Olympus BX61 photomicroscope with an Olympus DP70 digital camera (Olympus, Tokyo, Japan), equipped with an HBO 100 W mercury lamp and the corresponding filter sets for fluorescence microscopy: UV (365 nm, exciting filter UG-1), blue (450–490 nm, exciting filter BP 490), and green (545 nm, exciting filter BP 545). Photographs were processed using Adobe Photoshop CS software (Adobe Systems, San José, CA, USA).

### 2.11. Near Infrared (NIR) Imaging

For imaging experiments, the sample was placed in an optical setup especially designed for fluorescence imaging in the NIR-II, the PhotonSWIR Imager from BioSpace Lab (Nesles-la-Vallée, France). An InGaAs CCD camera (WiDy SenS 640V-ST, New Imaging Technologies, Paris, France) with a long-pass filter at 1000 nm (Thorlabs FEL1000, Newton, NJ, USA) and a fiber-coupled 808 nm laser diode (LIMO) were the principal components. The laser was set at a maximum output power of 6 W and collected by a lens for illumination of the samples with a maximum (peak) power density of 0.5 W/cm^2^ and an average power density of 17 mW/cm^2^. The system was operated in pulsed mode with 10 ms pulses (the resulting energy is 0.06 J) and a pulse-to-pulse separation of 35 ms (the pulse repetition rate is 28 Hz). A time delay between the end of the laser pulse and the camera acquisition was set to 20 µs at the beginning of the acquisition sequence. Then it was increased by 20 µs between each picture (exposure time 20 ms) and a series of 500 pictures was obtained, resulting in a final delay of 10 ms.

## 3. Results

NaGdF_4_: Nd^3+^, Yb^3+^, Tm^3+^ NPs were selected due to their prowess for in vivo imaging, as was already demonstrated [25]. Additionally, these NPs present an intense UC green emission that can be exploited for FRET-assisted excitation of EY. Figure 1B shows the emission spectrum of NaGdF_4_: Nd^3+^, Yb^3+^, Tm^3+^ together with the excitation and emission spectra of EY. A clear overlap between the ^4^G_7/2_ " ^4^I_9/2_ transition of Nd^3+^ and the absorption of EY can be observed. To prove the ability of transfer between the two systems, a solution containing UCNPs and EY was prepared and excited with a 790 nm laser. Figure 1C shows a clear reduction on the intensity of the Nd: ^4^G_7/2_ " ^4^I_9/2_ that also affects to the Nd: ^4^G_7/2_ " ^4^I_11/2_, as it originates from the same energy level (see energy scheme in Appendix A). The appearance of a red broad band is also observed, which is consistent with the EY emission. Therefore, we proceeded to the preparation of a system where the EY was linked to the UCNP (EY-UCNPs).

The NPs were synthesized according to the thermal decomposition procedure indicated in the experimental section. Transmission electron microscopy (TEM) images showed the formation of NPs with a diameter of 29.0 ± 0.4 nm (Figure 2A,B). The functionalization of the NPs with EY was performed via a nonspecific electrostatic interaction between the positively charged surface of the NPs and the negatively charged carboxylate group of EY. Initially, the oleate ligands that remained on the NPs surface were removed by re-suspending the NPs in an acidic aqueous dispersion, inducing the re-protonation of the oleate molecules, neutralizing them, and releasing ligand-free NPs, stable in aqueous solutions. Subsequently, EY was added to the dispersion, whose pH had been increased to 8–9, enabling the carboxylate of EY to bind to the surface of the NPs. After a vigorous stirring, modified EY-UCNPs were precipitated and thoroughly washed to remove the excess EY. The prepared EY-UCNPs showed limited dispersibility in water, probably because of the attachment of the carboxylate groups to the surface of the NPs, resulting in a loss of hydrophilicity. The functionalization process did not affect the morphology of the particles, as can be seen in Appendix A.

Hence, we also decided to explore the attachment of EY to the UCNPs by employing PEG molecules as a linker to improve the water dispersibility of the particles. The EY-PEG-UCNPs were prepared following the procedure described in the experimental section. The bifunctional PEG was bound electrostatically via its carboxylate to the positively charged surface of the NPs, while in parallel, EY was activated with typical EDC/NHS chemistry to facilitate the formation of the covalent amide bond between the amino group of the PEG ligand and the activated carboxyl group of EY. The attachment of the ligands to the surface resulted in an increase in the hydrodynamic diameter as shown in Figure 2C, going from 32 nm for ligand-free UCNPs, to 50 nm for EY-UCNPs, and 100 nm for the EY-PEG-UCNPs, with polydispersity indexes of 0.6, 1.2, and 0.4 respectively, where values superior to 0.7 indicate a polydisperse distribution of nanoparticles [26]. This increase could be related to the presence of the ligand covering the NPs’ surface and suggests that no big aggregation is occurring due to the functionalization.

Further confirmation of the presence of EY and PEG-EY was obtained from Fourier-Transformed Infrared (FTIR) measurements (Figure 2D). FTIR spectra revealed the subsequent replacement of the ligands on the NPs surface. The presence of oleate on the as-synthesized NPs is revealed by the alkene (C=C) stretching band at 2921 cm^−1^, and the bands at 1450 and 1560 cm^−1^, which have been previously assigned to the symmetric (δ_s_) and the asymmetric −COO− (δ_as_) stretches, respectively [23]. The disappearance of these bands after the acidic treatment indicates the successful removal of the oleate ligands. The FTIR spectra of EY-UCNPs and the EY-PEG-UCNPs present several features in the region between 1000 and 2000 cm^−1^ that can be assigned to EY’s xanthene-based vibrational modes [27]. An estimation of the EY concentration on the NPs surface was obtained by colorimetric studies, which indicated the presence of 4.34 · 10^−5^ mmol/mL and 2.89 · 10^−5^ mmol/mL mmol/mL of EY for EY-UCNPS and EY-PEG-UCNPs, respectively, as shown in Appendix A.

The emission spectra of both EY-UCNPs and EY-PEG-UCNPs under 790 nm excitation revealed a decrease in the green bands of the UCNPs (Figure 3A). However, the lack of emission of EY could be attributed to the already known concentration quenching of the EY molecules due to the high local concentration that exists on the surface of the UCNPs [18,28,29]. To confirm the presence of EY, both EY-UCNPs and EY-PEG-UCNPs were excited in a fluorimeter at 535 nm, and the corresponding emission band of EY could be detected, confirming the presence of the dye (Figure 3B). Moreover, the emission spectra did not show the intensity decrease and spectral shifts associated with the aggregation of EY. Therefore, we discarded the presence of EY aggregates and attributed the absence of the EY band under NIR excitation to the combination of low concentration of molecules and the decreased efficiency of the FRET process compared to direct excitation.

Since the main action of a PS is related to the production of singlet oxygen and/or other cytotoxic oxygen species generated under light irradiation, it is important to evaluate the toxicity of the PS in the absence of light. In fact, an important requirement for a compound to be considered a good PS for application in PDT is not to be toxic in the absence of light [30,31]. Therefore, the dark toxicity of the two types of EY conjugated NPs (EY-UCNPs and EY-PEG-UCNPs) and the initial UCNPs was evaluated. Experiments were carried out using two different NPs concentrations (0.1 and 0.02 mg/mL) that were administered to HeLa cell cultures. The cells were incubated in the dark for a period of 2 h and the cytotoxic activity of different EY conjugated UCNPs was evaluated using the MTT assay. As can be observed in Appendix A, the survival rates obtained show that UCNPs were nontoxic to HeLa cells in the absence of light at the two concentrations studied. In the case of both EY UCNPs conjugates (EY-UCNPs and EY-PEG-UCNPs), no loss in cell viability greater than 10% was detected at the two tested concentrations.

The efficiency of compounds that can act as photosensitizers in cell cultures and tumors, is directly related to their chemical structure, concentration, incubation time, light doses, and subcellular location. In this sense, it has been described that a wide range of cellular components may be targets of reactive, cytotoxic oxygen species formed during the photodynamic process. These cellular elements can be mitochondria, lysosomes, Golgi, and the plasma membrane. Therefore, we first evaluated the cell internalization and localization of both EY-UCNP conjugates. The subcellular localization of the EY-UCNP conjugates in HeLa cells was analyzed by fluorescence microscopy under UV (365 nm), blue (450–490 nm) and green (545 nm) light excitation (Figure 4).

The luminescent signals of EY-UCNPS and EY-PEG-UCNPs were observed after 2 h of incubation with a concentration of 0.1 mg/mL of both compounds. The corresponding control (without EY conjugate) showed a very low signal corresponding to mitochondrial autofluorescence under UV light (Figure 4, first row). Under blue emission, the cell cultured with EY-UCNPs and EY-PEG-UCNPs showed a green-yellowish signal homogenously distributed inside the cell with some brighter points that correspond to conjugates probably concentrated in cytoplasmic vesicles. We attribute this green fluorescence to the ^4^G_7/2_ → ^4^I_9/2_ emission of Nd^3+^ that can be excited under blue light through the conduction band of the NaGdF_4_ matrix. The yellowish color that is easily observed in the brighter points can be attributed to the mixture of the green emission of the NPs with the red emission of the EY excited by the green emission of the NPs. On the other hand, the red emission observed under green excitation is attributed to EY emission, which, according to the excitation and emission spectra shown in Figure 1B, can be directly excited with green light. Together, these results show that the two EY nano-conjugates are successfully located inside the cancer cells, which could optimize the efficiency of PDT.

The ability of EY-UCNPs and EY-PEG-UCNPs to generate reactive oxygen species (ROS) was tested by using the chemical trap 1,3-diphenylisobenzofuran (DPBF). A solution containing dispersions of EY-UCNPs (or EY-PEG- UCNPs) and DPBF in a mixture of DMSO and water was introduced in a light-isolated cuvette chamber and illuminated with the 790 nm diode laser. The absorption spectrum of the mixture was recorded at different irradiation times, as shown in Figure 5A. The band at 540 nm corresponds to EY, and the one at 410 nm corresponds to DPBF. After irradiation, the EY band remained mostly unchanged, while the one corresponding to DPBF decreased progressively with the irradiation time, as can be seen in Figure 5A for the EY-UCNPs and in Appendix A for the EY-PEG-UCNPs. Several control experiments were performed to ensure that the quenching of the DPBF band was related to the combined action of UCNPs, EY, and light, and not due to degradation of the compound or residual light entering the chamber. The different experiments were compared in terms of ROS generation efficiency (ROS_eff_), which was calculated as indicated in Equation (1).
(1)$$ROS_{eff}=\frac{\alpha -\alpha_{0}}{\alpha_{0}}$$
where *α* refers to the maximum of the molar attenuation coefficient of the absorption band of DPBF for a given irradiation time and *α*_0_ is the value before irradiation. The comparison between the *ROS_eff_* of the EY-UCNPs and EY-PEG-UCNPs with and without irradiation is shown in Figure 5B. Whereas in the case of the EY-PEG-UCNPs where the increase in ROS production in the presence of light is negligible compared to the dark control, the irradiation with NIR light of the EY-UCNPs induces a 4-fold enhancement on ROS generation, which demonstrates the ability of this system for PDT. Since this difference cannot be related to different EY concentrations between EY-UCNPs and EY-PEG-UCNPs, we attribute it to the longer distance that exists between the UCNPs and the EY, having a PEG linker molecule that results in the decrease in the FRET efficiency (which depends on the sixth power, R being^−6^, being R the distance between molecules). The concentration of produced ROS was obtained from the consumption of DPBF, which resulted to be 28 μM of ROS for the EY-UCNPs after 30 min of irradiation, comparable with other works based on commercial photosensitizers [32].

Then, a PDT study was performed by illuminating HeLa cells with a commercial NIR irradiation lamp. A set of long-pass filters was used to block lamp emissions under 750 nm, ensuring that no direct excitation of EY was occurring. The resulting spectrum of the lamp-filters system can be observed in Appendix A. The distance between the lamp and the cells was also adjusted, and an air-cooling system was used to eliminate the thermal component of the lamp until a negligible heating was observed in the cell culture. Cell viability was evaluated with HeLa cells using the MTT colorimetric assay [24] and cell viability data are summarized in Figure 6. Based on dark toxicity experiments (see Appendix A), a concentration of 0.1 mg/mL of NPs was employed. After incubating the cell cultures in the dark with both EY conjugates and UCNPs alone for 2 h, the cells were irradiated with the NIR lamp for 5, 15, and 30 min. Neither light-only treatment, nor UCNPs alone, resulted in a relevant photocytotoxic effect, since the maximum toxicity obtained was around 6% for cells treated with light. On the other hand, when the cells were treated with EY-UCNPs and illuminated for 5 min with the NIR lamp, approximately 10% phototoxicity was obtained. Irradiation for 15 min resulted in 40% toxicity. Further increases in irradiation time up to 30 min did not increase phototoxicity. In the case of EY-PEG-UCNPs, lower phototoxicity values of about 20% were obtained. These results were in agreement with the previous ROS generation experiments, which showed that the ROS generation efficiency of the EY-UCNPs was higher than EY-PEG-UCNPs. While cytotoxic effects produced by both EY conjugates in the presence or absence of light irradiation, there is a clear increase in cellular toxicity when the cells have been irradiated, confirming that the cytotoxic effect is due to reactive oxygen species produced under NIR light irradiation.

Finally, we ensured that the capability of our NPs for simultaneous deep tissue imaging in the NIR-II while being deployed as theragnostic agents was preserved. A series of intensity-based and fluorescence-lifetime-based imaging experiments are shown in Figure 7. First, EY-UCNPs and EY-PEG-UCNPs solutions were placed in a NIR imaging setup, illuminated by an 808 nm diode laser, and the image of the emission above 1000 nm was registered with a NIR camera.

The fluorescence image of both samples, and the merged figure of the optical and fluorescence images can be observed in Figure 7B,C. The fluorescence lifetime map, shown in Figure 7D, was calculated from a series of time-gated images, following the procedure described previously [33]. The lifetimes obtained are in good agreement with the fluorescence lifetime obtained from the time decay curves represented in Appendix A, measured by regular time resolved methods as described in the Experimental Section. The samples containing both EY nanoconjugates were covered with chicken muscular tissue (8 mm thickness) and the same images were acquired (Figure 7, second row (E-H)). The fluorescence from the samples can still be clearly seen through the tissue, evidencing that the EY-UCNPs and the EY-PEG-UCNPs preserve the imaging capabilities of the unfunctionalized UCNPs.

## 4. Conclusions

Two eosin Y (EY) nanoconjugates derived from UCNPs have been successfully prepared and tested as theragnostic nano-agents. The UCNPs demonstrated to be excellent EY nano-carries in living cells, resulting in their internalization, mediated PDT using 808 nm excitation, and allowed deep tissue bioimaging employing NIR-II emissions. The use of 800 nm excitation wavelengths ensured minimum cell damage, since the absorption of water is negligible, and increases tissue penetration enhancing the specificity of the treatment. Photophysical evaluations revealed better ROS production of directly functionalized EY-UCNPs versus EY-PEG-linked UCNPs. Gratifyingly, living cell studies disclosed excellent behavior of the UCNPs as both nanocarriers and energy transfer mediators. Thus, both EY nanoconjugates were capable of crossing the cell membrane of living HeLa cells, thereby making the internalization of the PS possible and allowing it to be excited under 800 nm irradiation light, bypassing the need of direct green light excitation and circumventing future autofluorescence issues. Moreover, the potential use of these nanoconjugates as theragnostic agents, for both PDT and diagnosis, was demonstrated combining PDT and fluorescence deep tissue imaging in the NIR-II

## Figures and Tables

**Figure 1 cancers-15-00102-f001:**
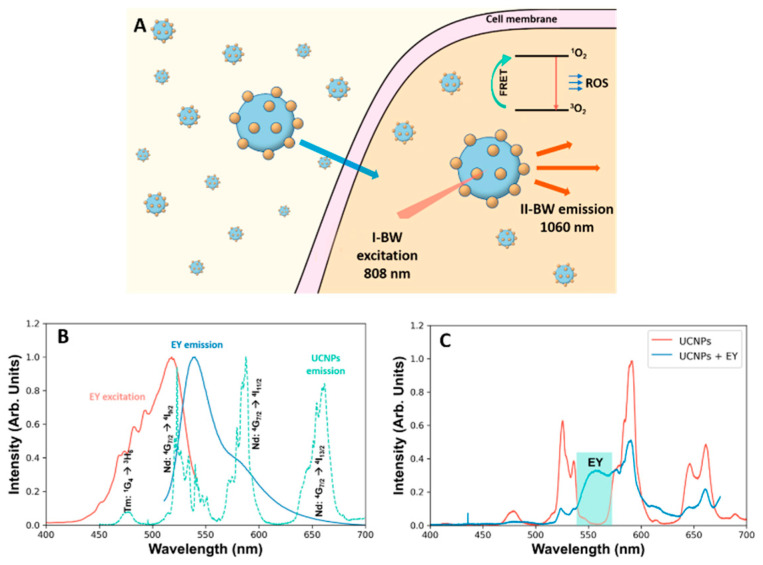
(**A**) Schematic drawing of the working process of EY-coated NPs at a cellular level. (**B**) Excitation spectrum (red) and emission spectrum (blue) of EY. The emission of the UCNPs is also included in the graph (green). (**C**) Emission spectra of a solution of UCNPs and a solution of UCNPs + EY where the quenching effect can be observed especially at the Nd: ^4^G_7/2_ " ^4^I_9/2_ and Nd: ^4^G_7/2_ " ^4^I_11/2_ transitions.

**Figure 2 cancers-15-00102-f002:**
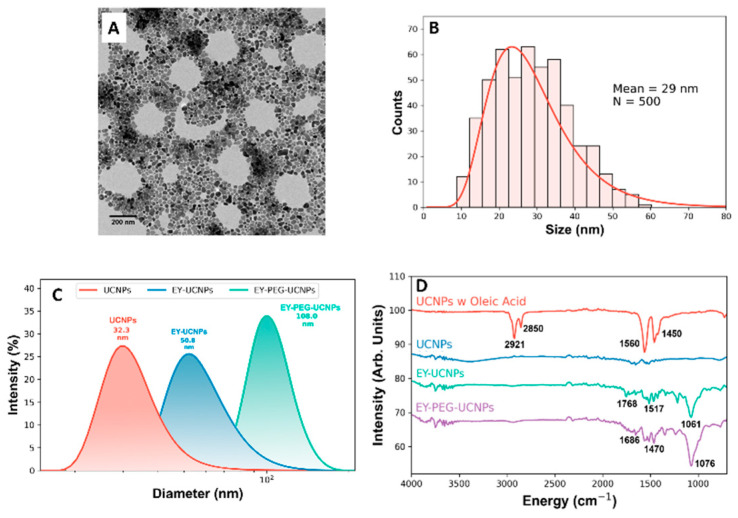
(**A**) TEM image of the UCNPs used in this work. (**B**) Particle size distribution histogram of the UCNPs. (**C**) Diameter distribution measured by DLS for UCNPs (red), EY-UCNPs (blue), and EY-PEG-UCNPs (green). (**D**) FTIR spectra of UCNPs, EY-UCNPs, and EY-PEG-UCNPs.

**Figure 3 cancers-15-00102-f003:**
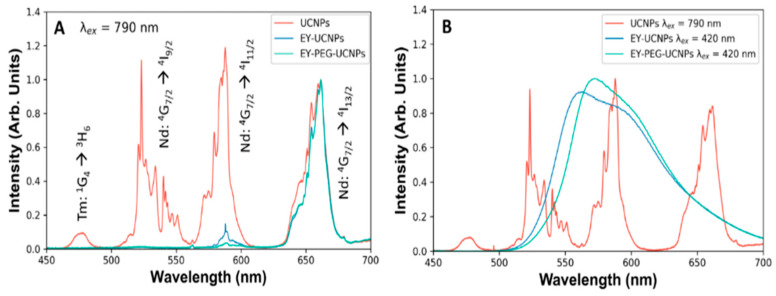
(**A**) Emission spectra of UCNPs, EY-UNCPs, and EY-PEG-UCNPs in the visible region. (**B**) Comparison between the emission bands of UCNPs and the emission bands of EY located in the EY-UCNPs and EY-PEG-UCNPs structures, under direct excitation of EY.

**Figure 4 cancers-15-00102-f004:**
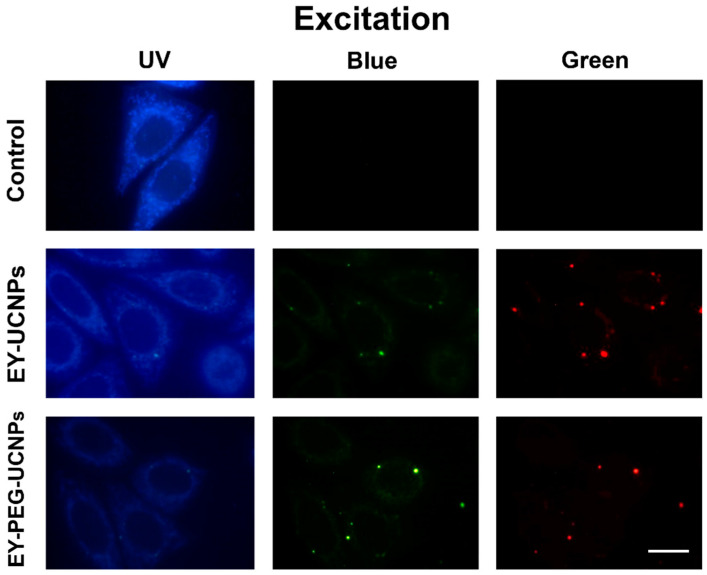
Subcellular localization of EY-UCNPs and EY-PEG-UCNPs in HeLa cells by fluorescence microscopy. Cells were incubated with EY-UCNPs conjugates and illuminated with UV, blue, and green light in the epifluorescence microscope. Scale bar: 10 µm.

**Figure 5 cancers-15-00102-f005:**
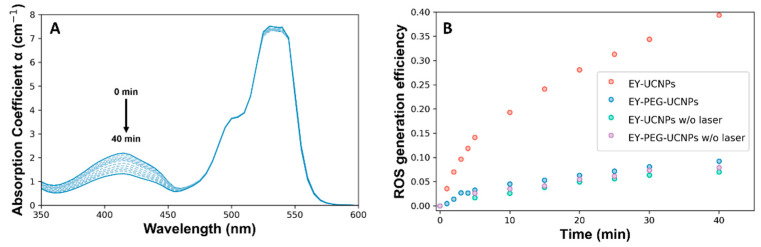
(**A**) Change in the absorption band after irradiation of nanoconjugate dispersions and DPBF with a 790 nm laser for 40 min. (**B**) ROS generation efficiency of EY-UCNPs and EY-PEG-UCNPs with (black and red) and without (yellow and green) 808 nm laser irradiation at different irradiation times.

**Figure 6 cancers-15-00102-f006:**
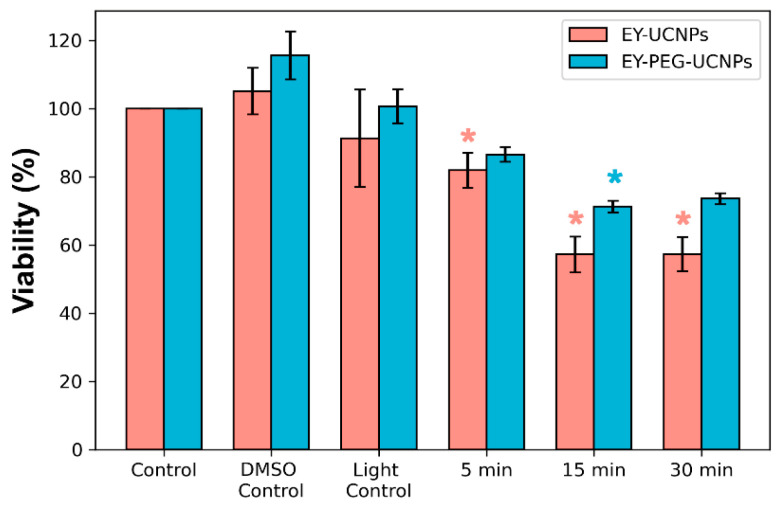
Viability of HeLa cells with EY-UCNPs and EY-PEG-UCNPs after 5, 15, and 30 min of irradiation with a NIR lamp. Both control experiments for DMSO and IR irradiation in the absence of UCNPs are also included. A statistical analysis was performed using a two-sample student’s *t*-test, and stars indicate significant differences between the control and the analyzed sample with *p* values < 0.05. (*) indicates 0.01 < *p* < 0.05.

**Figure 7 cancers-15-00102-f007:**
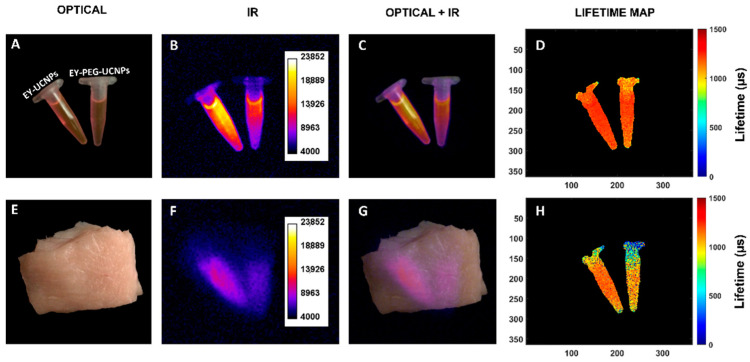
Intensity and lifetime fluorescence images of 1 mg/mL dispersions of EY-UCNPs and EY-PEG-UCNPs. First row: Visible image (**A**), NIR image (**B**), overlap between the optical and IR images (**C**) and lifetime maps (**D**) of EY-UCNPs and EY-PEG-UCNPs. Second row: visible image (**E**), NIR image (**F**), overlap between the optical and IR images (**G**) and lifetime maps (**H**) of EY-UCNPs and EY-PEG-UCNPs microtubes covered by 8 mm of chicken tissue.

## Data Availability

Data might be available through direct request to the authors.

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
