# Peer review of "Eosin Y-Functionalized Upconverting Nanoparticles: Nanophotosensitizers and Deep Tissue Bioimaging Agents for Simultaneous Therapeutic and Diagnostic Applications"

_cancers, 2022, doi:10.3390/cancers15010102_

Round 1

Reviewer 1 Report

The manuscript presents research to demonstrate that eosin EY nanoconjugates derived from UCNPs has a role of theragnostic nano-agent, and can use for both PDT and diagnosis.

Different modern techniques are used to evaluate this role, but some of them no too deeply described. There are beneficial results reported.  Figures are representative and in good quality. There are some errors identified which you have to resolve but could be much more.

1.Introduction: in this text "Biological evaluations, ..., revealed a unique behaviour of eosin UCNPs as a novel nanophotosensitizer ..." is a as a conclusion, not a target here.

2.Materials  ..

- unclear for me NaGdF4:Nd3+, Yb3+, Tm3+ or NaGdF4: Nd, Yb, Tm; in same NPs or different with each ions in part? You prepared just as ingle type of UNPS

-"Briefly, the UCNPs were dispersed in water (10 mg/mL) and the pH was reduced to 3-4 by adding dropwise 1 M HCl dropwise."

-in whole manuscript " mL" nu "ml".

-. "The aqueous phase was washed 3x with 5 mL of DCM, where is ???.

2.2. to summary!... and without : "help"!

-about Figure 7 , the information about E is miss.

- A suggesting of mechanism scheme could add a plus value of work.

-all references must be revised: e.g. references 22 and 23, same journal written different.

Author Response

Reviewer #1

1.Introduction: in this text "Biological evaluations, ..., revealed a unique behaviour of eosin UCNPs as a novel nanophotosensitizer ..." is a as a conclusion, not a target here.

We thank the reviewer for pointing out this aspect. We have modified the sentence accordingly and now it reads:

Here, we report the study of rare-earth doped NPs decorated with EY, as a NIR-excitable photosensitizer for PDT and bioimaging. These decorated UCNPs were easily prepared by a simple surface functionalization of UCNPs with EY. We performed bio-logical evaluations, including cellular uptake and toxicity assays, and evaluated the ability of eosin-UCNPs as novel photosensitizer for PDT and its capability to produce ROS under NIR-light irradiation.  Finally, the potential of UCNPs to simultaneously act as contrast agent for intensity-based and fluorescence lifetime-based deep-tissue imaging was also evaluated.

The changes can be found in page 2, last paragraph of the section “1. Introduction”

2.Materials  ..

- unclear for me NaGdF4:Nd3+, Yb3+, Tm3+ or NaGdF4: Nd, Yb, Tm; in same NPs or different with each ions in part? You prepared just as ingle type of UNPS

The notation that we have used is standard in several scientific fields as chemistry or materials physics, we acknowledge the comment of the reviewer and have clarified it considering the multidisciplinary scope of this journal. We hope that it is now clear that all the dopants (Nd, Yb and Tm) are integrated in the crystalline matrix of the nanoparticles, and in the two cases pointed out by the reviewer we are talking about the same type of nanoparticles: a matrix of NaGdF4 doped with ions of Nd3+, Yb3+ and Tm3+ that replace Gd3+ ions in the crystal structure. We have changed the notation in the text to NaGdF4: Nd3+, Yb3+, Tm3+ to enhance clarity for the readers. The change appears for the first time in page 3, first paragraph of the section “2.1 Synthesis of rare-earth-doped nanoparticles”.

-"Briefly, the UCNPs were dispersed in water (10 mg/mL) and the pH was reduced to 3-4 by adding dropwise 1 M HCl dropwise."

Thank you for pointing out this mistake, we have already corrected it (Page 3, at the end of the first paragraph of section “2.1 Synthesis of rare-earth-doped nanoparticles”.

-in whole manuscript " mL" nu "ml".

Thank you for noticing this, we have corrected the places where mL was incorrectly indicated.

-. "The aqueous phase was washed 3x with 5 mL of DCM, where is ???.

Thank you for pointing this out, we have corrected this mistake and added the full name of the chemical (dicloromethane) to the text (Page 3, at the end of the first paragraph of section “2.1 Synthesis of rare-earth-doped nanoparticles”.

2.2. to summary!... and without : "help"!

We have modified the text to remove help (Page 4, first paragraph, section “2.2 Synthesis of the NaGdF4: Nd3+,Tm3+,Yb3+@eosin Y NPs (EY-UCNPs)”, and the mechanisms appear explained in detail in the results section (page 6, second paragraph).

-about Figure 7 , the information about E is miss.

We have included the information about Figure 7.E, thank you for noticing this. The information can be consulted in page 11, last paragraph of the section “3. Results”.

- A suggesting of mechanism scheme could add a plus value of work.

In Figure 1.A we included a schematic of the working mechanism of our eosin-UCNPs structures. Our goal with this figure was to clarify that once the structures are internalized into the cells, the external excitation using NIR light will lead to the production of ROS inside the cells, leading to cell death. Thanks to the suggestion of the reviewer we now also included below’s additional scheme in the supporting information as Figure S1, which focuses on the photodynamic therapy part of the mechanism.

-all references must be revised: e.g. references 22 and 23, same journal written different.

We have carefully revised the references. These two particular references have almost the same name, but we have checked that they are in fact a two-part article:

Part 1: Eosin Y–Macromolecule Complexes. Part 1. —Application of Exciton Theory to the Study of the Arrangement of Eosin Y Molecules in Polycation-Induced Eosin Y Dimers

Part 2: Eosin Y–Macromolecule Complexes. Part 2. —Interactions between Eosin Y and Polycations, a Cationic Surfactant and Proteins

Reviewer 2 Report

The manuscript reported as “Eosin Y-Functionalized Upconverting Nanoparticles: Nanophotosensitizers and Deep Tissue Bioimaging Agents for Simultaneous Therapeutic and Diagnostic Applications” demonstrated near-infrared light absorbing functionalized upconversion nanoparticle loaded with Eosin-Y for deep penetration cellular imaging and photodynamic therapy of cancer. I highly recommend the manuscript for publication after going through a major revision. The author should go through the following points.

1.      The author should clearly explain the design chemistry of the final product. The reaction was done by simply mixing and stirring. I am afraid that it does not satisfy the reader's attention. What is the mechanism of interaction between upconversion NPs, PS, and PEG?

2.      TEM image of the material is not clear. Can the author provide TEM images of upconversion NPs before and after Eosin-Y conjugation? Please use a magnified resolution while taking the picture.

3.      Zeta potential values of the material are lower than 30 mV. On this basis, the authors should state that both nanomaterials possess poor colloidal stability in water. Is it a limitation to applying these platforms for PDT of cancer? Provide your material's polydispersity index (PDI) at every stage and discuss it accordingly.

4.      Figure 4 needs to be revised. I think the label of the column is wrong, like UV, blue and green. Need revision.

5.      Data presentation has to be taken care of at this level. Figure 5b is not consistently organized. The ROS generation effects of UCNP-PEG-Eosin-Y is deplorable. What is the difference between red and orange lines? The author should examine the problem in this section.

6.      Biocompatibility tests of the material should be carried out at different incubation times to evaluate the toxicity of PS without laser irradiation.

7.      The literature surveys do not significantly explain the current progress. References were bounded up to 2020. It should be revised to date. 

Round 2

Reviewer 2 Report

The manuscript can be accepted for publication now.

Appreciate the author for your response.